# Transparent dynamic infrared emissivity regulators

Yan Jia[1], Dongqing Liu ®[1] ✉, Desui Chen[2], Yizheng Jin ®[2], Chen Chen[1], Jundong Tao[1], Haifeng Cheng ®[1] ✉, Shen Zhou ®[1,3], Baizhang Cheng[1], Xinfei Wang[1], Zhen Meng[1] & Tianwen Liu[1]

Dynamic infrared emissivity regulators, which can efficiently modulate infrared radiation beyond vision, have emerged as an attractive technology in the energy and information fields. The realization of the independent modulation of visible and infrared spectra is a challenging and important task for the application of dynamic infrared emissivity regulators in the fields of smart thermal management and multispectral camouflage. Here, we demonstrate an electrically controlled infrared emissivity regulator that can achieve independent modulation of the infrared emissivity while maintaining a high visible transparency (84.7% at 400–760 nm). The regulators show high degree of emissivity regulation (0.51 at 3–5 μm, 0.41 at 7.5–13 μm), fast response (<600 ms), and long cycle life (>$10^4$ cycles). The infrared emissivity regulation is attributed to the modification of the carrier concentration in the surface depletion layer of aluminum-doped zinc oxide nanocrystals. This transparent infrared emissivity regulator provides opportunities for applications such as on-demand smart thermal management, multispectral displays, and adaptive camouflage.

Although infrared radiation is invisible to the human eye, all objects in our surroundings continually emit thermal infrared electromagnetic radiation. Recently, the development of technology for regulating dynamic infrared radiation has emerged as an attractive research area in the energy and information fields. The Stefan–Boltzmann law states that infrared radiation can be regulated by controlling either temperature or emissivity. Controlling the temperature requires devices with high energy consumption[1] and complex systems[2], while regulating the emissivity electrically is a promising method because of its flexible regulation, fast response speed, lightweight structure, and low-energy consumption[3–9]. Several electrically controlled dynamic infrared emissivity (DIE) regulators have been proposed based on ion intercalation/extraction into/from materials (e.g., metal oxides[3], conducting polymers[4], carbon nanomaterials[5,6]), electron injection/extraction into/from structures (e.g., quantum wells[7], plasmonic

resonators[10]), and reversible metal electrodeposition[11]. These existing electrically controlled DIE regulators are usually opaque owing to the strong absorption or reflection of visible light from the DIE materials or multilayer devices that are generally black (carbon nanomaterials[5,6]), white (lithium titanate[12]), or colored (e.g., devices based on polyaniline[4] or tungsten oxide[3]); see Supplementary Table 1 for details. The device opacity limits advanced applications with broad-spectrum requirements or multispectral compatibility. For smart thermal management of smart windows, it is necessary to simultaneously achieve independent regulation of solar spectral and infrared emissivity to better meet diverse thermal control requirements. For multispectral adaptive camouflage, the infrared emissivity and visible color should be dynamically regulated independently to counter multispectral reconnaissance in different spatial and temporal scenarios and achieve both visible and infrared chameleon-like camouflages. Highly

---

[1]Science and Technology on Advanced Ceramic Fibers and Composites Laboratory, College of Aerospace Science and Engineering, National University of Defense Technology, Changsha, P.R. China. [2]Key Laboratory of Excited-State Materials of Zhejiang Province, State Key Laboratory of Silicon Materials, Department of Chemistry, Zhejiang University, Hangzhou, P.R. China. [3]Institute for Quantum Science and Technology, College of Science, National University of Defense Technology, Changsha, P.R. China. ✉e-mail: liudongqing07@nudt.edu.cn; hfcheng@rocketmail.com

transparent DIE regulators are anticipated as they can be placed on top of a solar spectrum dynamic regulation device or visible color-changing device to achieve independent dynamic regulation of the infrared emissivity and solar spectrum. Wang et al.[13] and Sun et al.[14] prepared DIE regulators based on $VO_2$ with visible transparencies of 27.8% (380–780 nm) and 62% (400–780 nm), respectively. However, the passive thermochromic emissivity regulation characteristics and high transition temperature (~60 °C) of $VO_2$ have limited the application of these regulators.

Electrical manipulation localized surface plasmon resonance (LSPR) of transparent conductive oxide nanocrystals (NCs) in the near-infrared band (0.78–2.5 μm) has been demonstrated and used to regulate the solar spectrum in smart windows[15,16]. In contrast to the regulation of the near-infrared spectrum (accounting for ~50% of the solar spectral energy), which is usually used for the modulation of the solar spectrum energy, DIE regulation of the mid-infrared region (2.5–25 μm) can regulate the outward radiated energy of an object itself. However, the application of LSPR regulation in DIE regulator is limited due to the infrared opacity of substrate and infrared reflection of the electrode shielding the regulation of LSPR in the mid-infrared band.

Here, we propose a DIE regulation mechanism and design a fully transparent DIE (TDIE) regulator based on aluminum-doped zinc oxide (AZO) NCs by the electro-regulation of LSPR at infrared wavelengths. The LSPR absorption intensity of the AZO NCs is increased or decreased by modulating the carrier concentration in the surface depletion layer of the AZO NCs. The capacitive characteristics of electron injection/extraction in the LSPR regulation process make it possible for the working electrode to be placed under the AZO NC layer, which avoids shielding infrared radiation and provides an infrared reflection background for LSPR infrared regulation. The developed TDIE regulators exhibit a DIE regulation of 0.51 and 0.41 at the mid-wave infrared (MWIR; 3–5 μm) and long-wave infrared (LWIR; 7.5–13 μm) atmospheric windows, respectively, and the visible trans-

mittance is maintained at 84.7%. Meanwhile, TDIE regulators have a fast response time (<600 ms) and long cycle life ($10^4$ cycles). The TDIE regulators exhibit significant advantages for smart thermal management and multispectral display applications compared to state-of-the-art devices.

## Results

### Device structure and operating principle of TDIE regulators

By precisely controlling the Al dopant content in the AZO NCs (diameter: $11.0 \pm 0.6$ nm) to 0.95at%, LSPR absorption peak position was modulated to infrared wavelength of 7.58 μm (Supplementary Figs. 1 and 2 and Supplementary Note 2)[17]. The full-width half-maximum of the LSPR peak covers the entire infrared atmospheric window owing to free carrier motion scattering[18], allowing broadband regulation by the TDIE regulator (Supplementary Fig. 2). The structure and operating principle of the TDIE regulators are shown in Fig. 1a. $BaF_2$ was placed on top of the TDIE regulator as the visible- and infrared-transparent substrate, with transmittance ($T$) values of $T_{400–760\ nm} = 92.9\%$ and $T_{2.5–13\ μm} = 96.7\%$, respectively. An upper AZO NC film (~1.14 μm thick; Fig. 1a and Supplementary Fig. 3) was used to achieve variations in the LSPR intensity in the infrared wavelengths. The upper indium tin oxide (ITO) film (~330 nm thick; Fig. 1a and Supplementary Fig. 3) is the key layer for achieving DIE regulation and provides three functions. First, the upper ITO film acts as a transparent working electrode to provide transparency and the capacitive performance of the TDIE regulator. Second, it serves as an infrared reflectivity layer (with a reflectivity $R_{3–13μm} = 86.9\%$). If the infrared reflectivity layer is not present, the change in the LSPR absorption intensity of the AZO NC cannot be observed due to the high infrared absorption of the electrolyte. Third, as a medium for transmitting electrons, it injects/extracts electrons into/from AZO NCs. The lower AZO NC layer and ITO film act as the ion storage layer and transparent counter electrode, respectively. Benefiting from the high visible transmittance of each film, the $T_{400–760\ nm}$ values of $BaF_2$/AZO NC/ITO and AZO NC/ITO-glass half-devices were

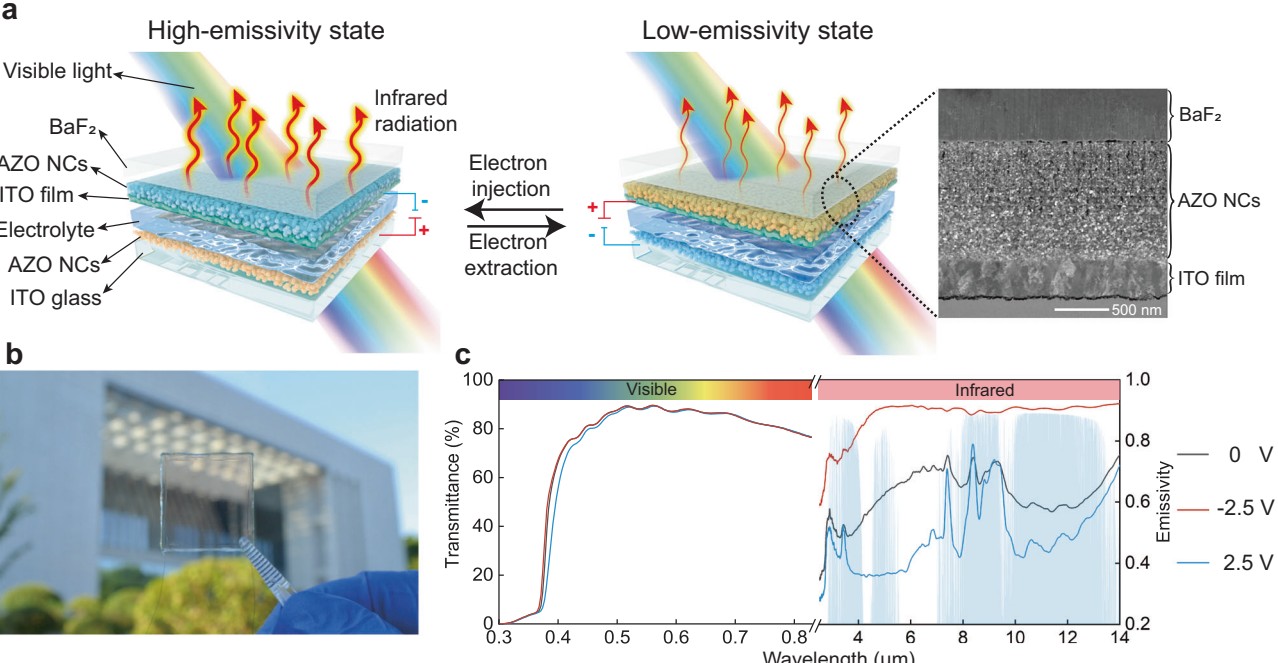

**Fig. 1 | Device structure and optical properties of the developed TDIE regulators. a** Schematic of the TDIE regulator structure. The device can be switched between a high-emissivity state (electron injection into AZO NCs) and a low-emissivity state (electron extraction from AZO NCs), where AZO NCs refer to aluminum-doped zinc oxide nanocrystals. Inset: morphology of the $BaF_2$/AZO NC/ITO (indium tin oxide) half-device. **b** Photograph of a TDIE regulator. **c** Visible transmittance and infrared emissivity spectra of TDIE regulators at various applied voltages; blue-shaded areas are infrared atmospheric windows.

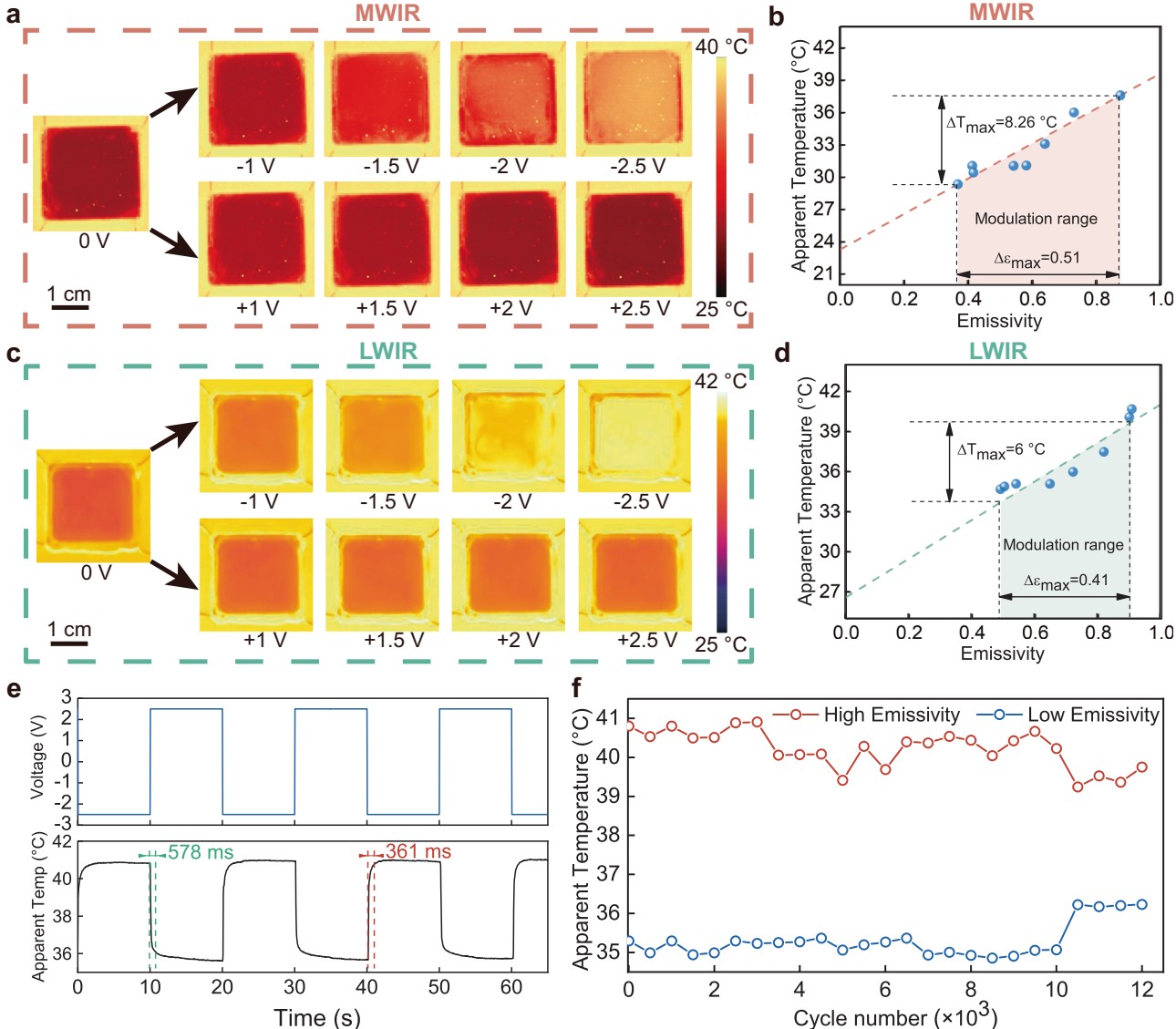

**Fig. 2 | Performance of the TDIE regulators. a** MWIR (mid-wave infrared; 3–5 μm) images of TDIE regulators at different voltages. **b** Regulation range of MWIR. **c** LWIR (long-wave infrared; 7.5–13 μm) images of TDIE regulators at different voltages. **d** Regulation range of LWIR. **e** Transient response of TDIE regulators during cycling. The response time is defined as the time elapsed between the application of the electrical signal and the temperature response reaching 90% of the final value. **f** High stability of the maximum and minimum apparent temperatures of LWIR during cycling, indicating the good endurance of the TDIE regulators. The test was conducted in the indoor laboratory environment.

81.2% and 88%, respectively (Supplementary Fig. 4). Finally, the visible transparency of a TDIE regulator was increased to 84.7% (Fig. 1b, c) using a transparent liquid LiTFSI/tetraglyme electrolyte with a refractive index ($n \approx 1.43$) matching that of the ITO film ($n \approx 1.394$) and AZO NCs ($n \approx 1.7$). The haze of the TDIE regulator is small, only 2.11%, due to the low scattering of the AZO NC film and the wetting effect of the liquid electrolyte (Supplementary Fig. 5). Moreover, the TDIE regulators maintained a high visible transmittance in both high- and low-emissivity states (Supplementary Fig. 7 and Supplementary Movie S1).

When the TDIE regulator is charged, electrons are injected into the AZO NCs, increasing the LSPR absorption and resulting in a high-emissivity state of the regulator (Fig. 1a, c). Conversely, when the TDIE regulator is discharged, electrons are extracted from the AZO NCs, LSPR absorption is reduced, and the transmittance increases. Thus, the TDIE regulator exhibits the high infrared reflectivity of the upper ITO film, that is, the low-emissivity state (Fig. 1a, c; Eq. (S3)). The infrared absorption peaks in Fig. 1c are attributed to the infrared absorption of the LiTFSI/tetraglyme electrolyte in the fingerprint region. The

existence of the absorption peaks is due to the presence of cracks in the ITO film, allowing the electrolyte to partially penetrate.

## Performance of TDIE regulators
The emissivity of the TDIE regulators is regulated by the applied voltage (Fig. 2a, c). The TDIE regulators exhibited the maximum emissivity regulation ($\Delta\varepsilon_{MWIR} = 0.51$, $\Delta\varepsilon_{LWIR} = 0.41$) with a charge capacity of $0.27 \pm 0.5$ mC/cm² (Fig. 2b, d and Supplementary Fig. 8). Emissivity regulation is not binary, and the intermediate state of the infrared emissivity can be precisely regulated by applying different voltage (Fig. 2a, c and Supplementary Fig. 9). With the change of the applied voltage, the number of electrons injected into the AZO NCs varies, resulting in the change of the LSPR absorption of the AZO NCs. For example, when a more negative voltage is applied, more electrons are injected into the AZO NC, and its LSPR absorption is stronger. Therefore, TDIE regulator exhibited different infrared emissivity after applying different voltage. The emissivity of the TDIE regulator can remain stable for more than 6 hours under different voltages

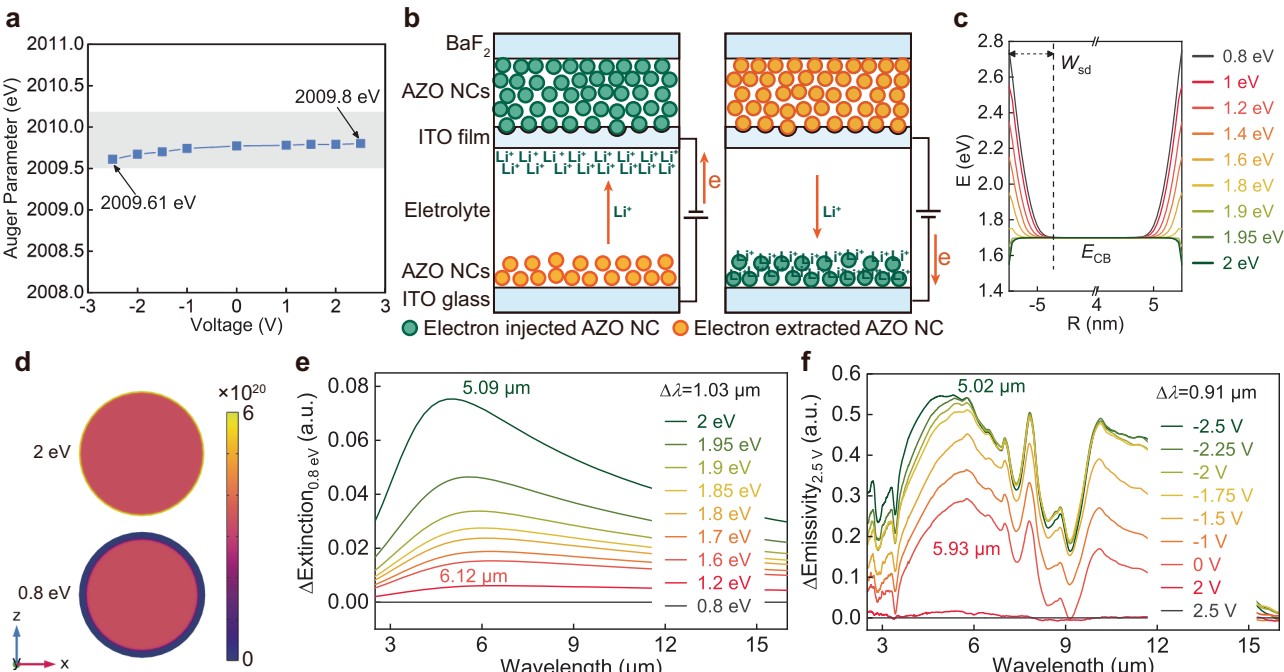

**Fig. 3 | Infrared regulation mechanism of the developed TDIE regulators.**
**a** Auger parameter of Zn in AZO NCs during the application of voltage from −2.5 to 2.5 V. The gray box shows the range of Auger parameters of ZnO. **b** Schematic of Li⁺ ion balancing charge in TDIE regulators. **c** Conduction band ($E_{CB}$) bending profiles of AZO NCs calculated at various applied surface potentials ($E_{surf}$) relative to the bandgap center (reference potential, $E_{ref}$). $W_{sd}$ refer to the width of the surface depletion layer. **d** Schematic of the surface depletion layer and radial carrier concentration ($n_e$) in AZO NCs at 0.8 eV and 2 eV. **e** Simulated spectra of the extinction change of an AZO film at various surface potentials relative to that at $E_{surf} = 0.8$ eV. The positions of the modulated peaks are marked. **f** Experimental spectra of the emissivity change of AZO films at various voltages relative to that at 2.5 V. $E_{ref}$ is the potential of the counter electrode (ITO glass).

(Supplementary Fig. 10). The memory time at 2.5 V ($\tau_{m+}$ = 1.29 s) was significantly shorter than that at −2.5 V ($\tau_{m-}$ = 1351 s; Supplementary Fig. 11), because the carrier depletion layer in the AZO NCs due to the positive voltage was rapidly compensated by free carriers from the ITO film[19]. The TDIE regulators exhibited a refresh rate of ~2 Hz (response times of 578 and 361 ms; Fig. 2e).

The cycling stability is a key indicator for DIE regulators. The developed TDIE regulators exhibited a cycle stability of >10,000 cycles (Fig. 2f). Moreover, the visible transmittance of the TDIE regulator was maintained at 83.4% even after 10,000 cycles (Supplementary Fig. 13). The TDIE regulators exhibited high visible transmittance (>80%); to the best of our knowledge, this has never been reported for previous electrically controlled DIE regulators (Supplementary Table 1).

Long-term outdoor operation stability is crucial for the application of TDIE regulators. TDIE regulator can operate stably under outdoor environment for over 6 days, shown in supplementary Note 4. During long-term outdoor operation of the TDIE regulator, the visible light transmittance of the TDIE regulator decreased to 58.9%, and the AZO nanocrystal film was partially corroded and peeled off due to the corrosion and oxidation of the electrolyte (Supplementary Figs. 19 and 20). More stable and less corrosive electrolytes will be adopted in future research work. We investigated the impact of environmental factors on the performance of TDIE regulators. The air humidity has no significant impact on the TDIE regulator. If dust and rainfall on the TDIE regulator, the TDIE regulator will show high-emissivity characteristics of dust and water (Supplementary Figs. 21 and 22).

**Regulation mechanism of TDIE regulators**
Understanding the underlying DIE regulatory mechanism of AZO NCs is important. The electrochromic properties of transition-metal oxides usually arise from surface redox reactions, ionic intercalation, and/or capacitive charging[20]. The electrochemical response of NCs can be used to distinguish between surface redox and capacitive charging[21].

Rectangular cyclic voltammograms were measured for the AZO NCs (Supplementary Fig. 23), and were similar to that of a capacitor, indicating that the electrochemistry is mainly dominated by capacitive charging[21]. With increasing voltage, the Auger parameter of the Zn element in AZO NCs slightly increased (Fig. 3a), but remained within the Zn(II)O range[22,23], indicating that no surface redox reaction occurred in the AZO NCs. This was also demonstrated by the lack of significant redox peaks in the cyclic voltammograms. Tetrabutylammonium (TBA⁺)-based electrolytes were used to determine if the DIE regulation of AZO NCs is triggered by ionic intercalation. Ionic intercalation into AZO NCs is not permitted because TBA⁺ (ionic diameter: 0.494 nm) is significantly larger than the crystal spacing of the AZO NCs (0.261 nm)[24,25]. The AZO films exhibited the same DIE regulation properties in the TBA⁺-based electrolytes and Li⁺-based electrolytes, indicating that the DIE regulation of AZO NCs is not due to ionic intercalation (Supplementary Fig. 25). Therefore, capacitive charging/discharging is the dominant mechanism for the DIE regulation of AZO NCs. Moreover, owing to their capacitive characteristics, the response time of the TDIE regulator was of the order of milliseconds[21].

The diffusion of Li⁺ ions in the electrolyte under positive or negative applied voltages was investigated. Li⁺ ions pass through the porous working electrode film to balance the charge in some electrically controlled DIE regulators with an infrared reflectivity layer as the working electrode[4,12]. Gold electrode is a commonly used blocking electrode to limit the transport of ion[26,27]. We used a dense gold film (~500 nm thick; Supplementary Fig. 26) as a working electrode instead of the ITO working electrode in the TDIE regulator because it is difficult for Li⁺ ions to pass through it[28,29]. DIE regulation was also observed in devices with a gold film as the working electrode (Supplementary Fig. 26). It indicated that Li⁺ ions do not pass through the working electrode in TDIE regulators and equilibrate the charge only on the surface of the working electrode when a negative voltage is applied

(Fig. 3b). When a positive voltage is applied to the working electrode, $Li^+$ ions equilibrate the charge in the ion storage layer (Fig. 3b).

The intensity modulation of the LSPR is primarily responsible for the DIE regulation of AZO NCs, which differs from the behavior of conventional electrochromic NCs caused by shifts in the LSPR peak position[15,16,30]. The modulation of the LSPR intensity and peak position can be explained by the formation of a dynamic depletion layer near the AZO NC surface. The surface depletion layer of a semiconductor NC is defined as the region where the carrier concentration differs significantly from that in the interior. The depletion layer is formed because the Fermi level is pinned to the surface potential by the natural surface state or applied voltage[19,31].

The carrier concentration distribution and energy band profiles of the AZO NCs were calculated at various applied surface potentials ($E_{surf}$) using Poisson's equation (Supplementary Note 5). The AZO NC energy band bends only in the surface layer of the carrier-depleted region (Fig. 3c). The surface potential was set from 0.8 to 2.0 eV, referring to previous literature[19]. Higher potentials above 2.0 eV were not set, since the result of solving Poisson's equation did not converge. When the oxidation potential ($E_{surf}$ = 0.8 eV) is applied, electrons were extracted from AZO NC (Supplementary Fig. 27). The width of the surface depletion layer ($W_{sd}$) of an AZO NC is -1.27 nm, occupying 54.3% of the NC volume (Fig. 3d). Electrons were injected into the AZO NC as the surface potential increased, resulting in a decrease in $W_{sd}$ (Supplementary Fig. 27). Under a 2 eV surface potential, electrons were injected to a depth of 0.715 nm from the surface of a typical AZO NC. The carrier concentration gradient did not penetrate deeply into the core of the NC (Fig. 3d and Supplementary Fig. 29).

The absorption spectra of different surface depletion layers, simulated for an individual AZO NC with carrier concentration variations confined to the shell of the NC, show that the intensity of the LSPR is modulated substantially while the peak position of the LSPR is slightly regulated ($\Delta\lambda$ = 0.89 μm; Supplementary Fig. 31). We further simulated the effective dielectric function of the AZO NC film using the Maxwell–Garnett effective medium model (Supplementary Note 5). The LSPR modulation of AZO NC films at different surface potentials relative to that at $E_{surf}$ = 0.8 eV was fitted (Fig. 3e). The LSPR intensity was modulated, accompanied by a slight peak position modulation ($\Delta\lambda$ = 1.03 μm; Fig. 3e). Notably, the LSPR modulation of the intensity and peak position predicted by the model (Fig. 3e) was in agreement with the observed experimental values (Fig. 3f). Therefore, the surface depletion layer is critical for the electrochemical DIE regulation of AZO NCs.

## Application demonstration of TDIE regulators

The high visible transparency of the developed TDIE regulators provides the opportunity for the independent modulation of visible and infrared spectra, making this technology suitable for several application scenarios. TDIE regulator can achieve a temperature regulation of 0.61 °C by regulating the emissivity of the TDIE regulator (Supplementary Fig. 34). We tested the radiated power regulation of the TDIE regulator outdoors (Supplementary Note 6). Compared with the radiative coolers[32–35], the TDIE regulator can achieve an infrared radiated power regulation of 23.1 W/m$^2$ (Supplementary Fig. 38). The experimental infrared radiated power regulation amount is lower than the calculated value (64.9 W/m$^2$ calculated by equation S5 and S6). This is because the calculated radiated power solely relies on the emissivity spectrum of TDIE regulator, whereas experimental radiated power regulation is affected by heat conduction and convection. Smart energy-saving (SES) buildings are designed by combining the use of SES roofs and SES windows, which are prepared by covering TDIE regulators on White/Black devices and smart glass, respectively (Fig. 4a). As illustrated in Supplementary Fig. 39, SES roofs allow for the independent modulation of visible reflectance (white: high reflectance

or black: high absorption) and infrared emissivity (high or low emissivity). SES windows enable the independent control of visible transmittance (transparent or blue) and infrared emissivity (high or low emissivity; Supplementary Fig. 40). Unlike passive radiative cooling materials[36,37] and temperature-adaptive radiative materials[13,38], which can only achieve radiative cooling or passive radiation modulation that is dependent of temperature, SES roofs and SES windows can operate in various modes, enabling on-demand thermal control and meeting building energy-efficiency requirements (Supplementary Figs. 41 and 42 and Supplementary Movie S2).

We simulated the energy consumption of SES buildings for heating, ventilation, and air conditioning (HVAC) using *EnergyPlus* software. For more details, see Supplementary Note 7. A Medium Office prototype building model from the U.S. Department of Energy with a window-to-wall ratio of 33% and a total floor area of 4982.19 m$^2$ was applied[39]. The Köppen climate classification divides the world into 30 climate zones, represented by typical cities in different climate zones[40]. For example, in Beijing, China, SES buildings can save 204 MBtu of the average annual HVAC energy with an energy savings of 12.3% compared with standard buildings (Supplementary Fig. 43), resulting in a 59,625 kg reduction in $CO_2$ emissions (Eq. (S17)). As shown in Fig. 4b, SES buildings can yield the highest HVAC energy savings in colder climate zones with energy savings of up to 464 MBtu per year, which is equivalent to a 111,076 kg reduction in $CO_2$ emissions. It is worth noting that both SES roofs and windows require only low driving voltages (<3 V) and have low power consumption. The driving energy of the TDIE regulator is only 5.87–7.87 J/m$^2$ (Supplementary Fig. 8). The energy consumed by SES roof and window is negligible compared to the amount of HVAC energy saved by the SES buildings. Furthermore, the potential of the TDIE regulator for smart thermal management in spacecraft is demonstrated in Supplementary Note 8.

TDIE regulators can also be applied in multispectral displays and adaptive camouflage in the visible and infrared wavelength regions. Conventional optoelectronic displays, such as organic and quantum-dot light-emitting diodes (LEDs), are designed to operate with high efficiency in a certain wavelength range, making it difficult to achieve multispectral displays[6]. We prepared three-by-three arrays in which TDIE regulators served as independently addressable "pixels" and infrared animations were realized using an electrical circuit and programming (Supplementary Note 9). In principle, the pixel density of the infrared multiplexed array can be further increased by reducing the size of a single TDIE regulator. A multispectral display that can display images in the visible and infrared wavelength regions simultaneously was realized by placing the multiplexed array on the LED display (Fig. 4c and Supplementary Movie S3). The multispectral display exhibits multi-channel independent information dissemination, fast refresh rate, and low power consumption. We also realized adaptive visible-infrared compatible camouflage by applying the TDIE regulator on an electrochromic device, which changed its visible and infrared characteristics depending on the background environment (Supplementary Note 10 and Supplementary Fig. 47).

## Discussion

In summary, we developed a DIE regulator with high visible transparency that provides an opportunity for the independent modulation of visible and infrared spectra. This functionality is expected to inspire more applications, either as a standalone unit or incorporated with an established visible light-manipulation device. Furthermore, the infrared plasmonic regulation in the NCs could enable the development for active plasmonics[41], transparent electronics[42], and other technologies based on infrared radiation. In the future, we expect this technology to be applied in a broad range with the development of flexible and large-area TDIE regulators.

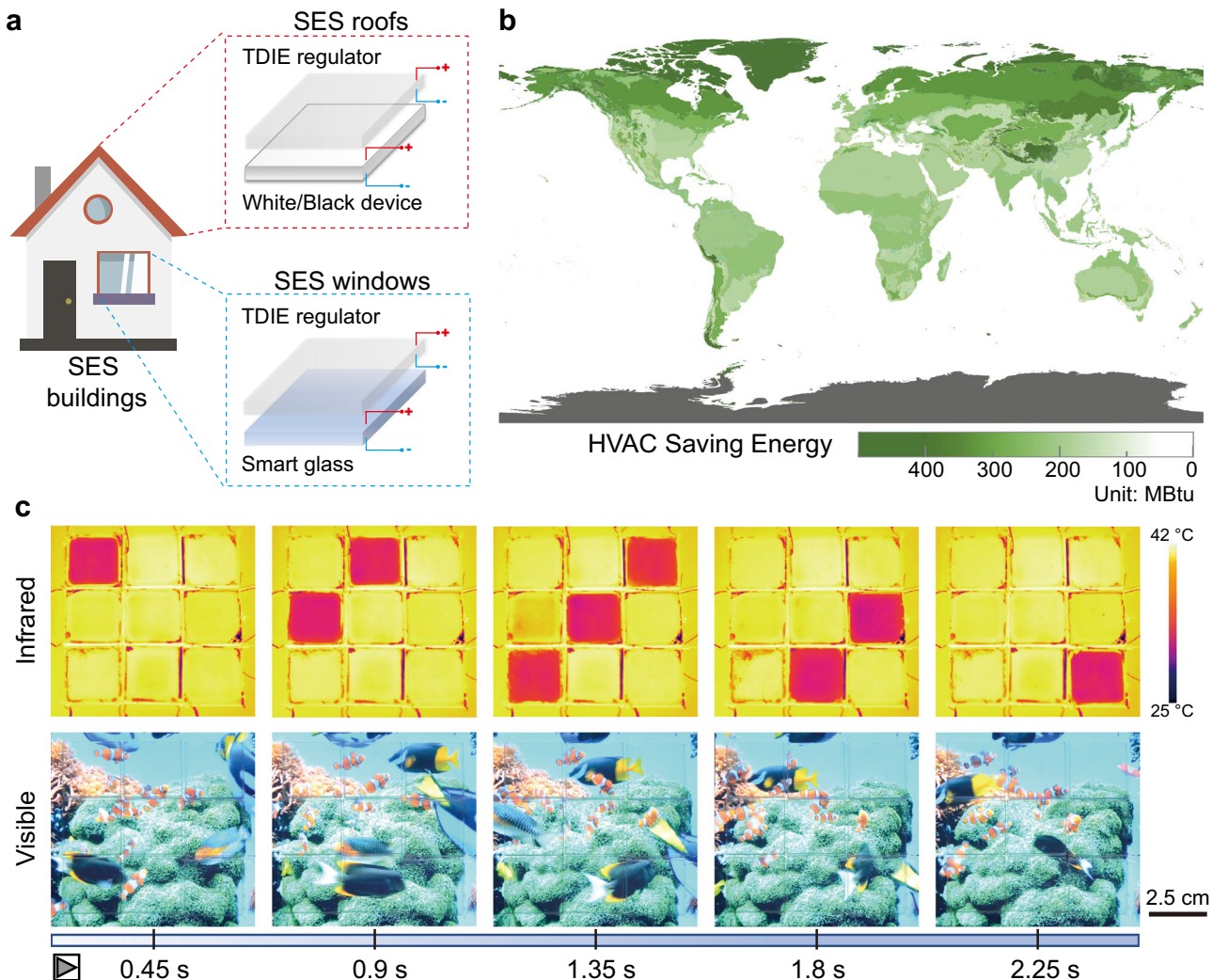

**Fig. 4 | Demonstration of TDIE regulator applications. a** Schematic of SES (smart energy-saving) roofs and windows in SES buildings. **b** Map of average annual HVAC (heating, ventilation, and air conditioning) energy savings achieved by SES buildings in different climate zones around the world compared to a standard building as the baseline. The map is licensed under the CC BY 4.0[40]. **c** Demonstration of a multispectral display. Visible and infrared movies are displayed simultaneously. The visible movie is created by LisaRedfern via *Pixabay*.

## Methods

### Chemical reagents

Zinc acetate dihydrate [$Zn(CH_3COO)_2 \cdot 2H_2O$ 99%], oleic acid (OA, 90%), and oleylamine (OLA, 70%) were supplied by Sigma-Aldrich. Aluminum acetylacetonate [$Al(Acac)_3$, 99%] and oleyl alcohol (80–85%) were supplied by Alfa Aesar. Toluene was purchased from Sinopharm Co., Ltd. Tin-doped indium oxide (ITO) pellets (In:Sn =90:10, 99.99%) were purchased from ZhongNuo Advanced Material Technology Co., Ltd. as an evaporation material. Commercial ITO glass (25 mm × 25 mm × 1.1 mm; ≤15 Ω) was purchased from Kaivo Optoelectronic Technology Co., Ltd.

Lithium bis(trifluoromethanesulfonyl)imide (LiTFSI, 98%), tetrabutylammonium perchlorate (TBAP, 98%), tetraethylene glycol dimethyl ether (98%), acetonitrile (98%), and propylene carbonate (PC, anhydrous, 99.5%) were purchased from J&K Scientific.

### Fabrication of TDIE regulators

The AZO NCs were synthesized using a slightly modified process as previously explained by Wainer et al.[15]. In a typical synthesis, 2.376 mmol of $Zn(CH_3COO)_2 \cdot 2H_2O$, 0.024 mmol of $Al(Acac)_3$, and 2.4 mL of OA were magnetically stirred under vacuum at 110 °C for 20 min. Then, 2.4 mL of OLA was injected. The precursors were further stirred under vacuum at 100 °C for 5 min and then kept at this temperature under $N_2$ atmosphere for further use. Meantime, 16 mL of oleyl alcohol was heated to 280 °C under a flowing nitrogen atmosphere. Then, 2.4 mL of the precursor solution was slowly injected into hot oleyl alcohol at a rate of 0.15 mL/min. After injection, the precursor was reacted for 1 h. Then, the reaction mixture was cooled to room temperature in an $N_2$ atmosphere. In total, 20 mL of ethanol was added to the reaction mixture, and the AZO NCs were separated after centrifugation at 9000 rpm for 5 min. Toluene was used to disperse the AZO NCs at a concentration of ~50 mg/mL.

AZO NCs were spin-coated onto 25 × 25 × 1 mm $BaF_2$ sheets at a spinning speed of 1000 rpm for 60 s, followed by 4000 rpm for 20 s. The spin-coating process was repeated several times to increase the film thickness to ~1.1 μm. The samples were then heated in an Ar environment at 250 °C for 30 min. An ITO film with a thickness of ~330 nm was then evaporated (MEB-600, Beijing Chuangshiweina Technology Co., Ltd., China) onto the samples as the working electrode at a deposition rate of 1 Å/s. The substrate temperature was set to 300 °C, and the oxygen flow rate was 20 standard cubic centimeters per minute (sccm).

The AZO NCs were spin-coated as an ion storage layer on the counter electrode ITO glass (25 × 25 × 1.1 mm; ≤15 Ω) at a spinning

speed of 1000 rpm for 60 s, followed by 4000 rpm for 20 s. The spin-coating process was repeated several times to increase the film thickness to ~500 nm. Half-devices made of $BaF_2$/AZO NC film/ITO film and AZO NC film/ITO glass were encapsulated using a transparent silicone adhesive sealant with 0.1-mm Ag wire as the lead. LiTFSI (1 M) in tetraglyme was used as the electrolyte. The electrolyte was injected into the device. The TDIE regulator was prepared according to the steps outlined above.

### Fabrication of smart glass and SES windows

$WO_3$ films (~400 nm) were deposited on an ITO glass (25 × 25 × 1.1 mm; ≤15 Ω) as an electrochromic layer using a high-vacuum four-target magnetron sputtering system (MSP-300BTI, Beijing Chuangshiweina Technology Co., Ltd., China). The deposition was performed at a pressure of 1 Pa and power of 120 W for 700 s. The Ar flow rate was set to 32 sccm and $O_2$ flow rate was set to 8 sccm. Half-devices made of ITO glass/$WO_3$ film and ITO glass were encapsulated using a transparent silicone adhesive sealant with a 0.1-mm Ag wire as the lead. The smart glass was prepared after injecting the electrolyte (1 M LiTFSI in tetraglyme). SES windows were assembled by stacking the TDIE regulators on top of the smart glass.

### Fabrication of white/black devices and SES roofs samples

SES roofs were built by stacking TDIE regulators on top of the White/Black device. The fabrication process for White/Black devices proposed by Imamura[43] was followed with a slight modification. A transparent silicone adhesive sealant was used as the encapsulation material instead of the Teflon spacer, and a 0.1-mm Ag wire was used as the lead.

### Fabrication of visible-transparent infrared displays

Nine TDIE regulators in a 3 × 3 array were arranged on an LED display (iPad mini 5). The wires were then extended and attached to the circuit. Details of the circuit and programming are presented in Supplementary Note 9.

### Fabrication of adaptive visible-infrared compatible camouflage device

A visible-infrared-compatible camouflage device was assembled by stacking a TDIE regulator on the electrochromic device. The preparation of the electrochromic device was previously described in ref. 44.

### Characterization

The total infrared reflectance spectra of the samples (2.5–25 μm) were measured using a Fourier transform infrared (FTIR) spectrometer (Bruker Vertex 70) equipped with a mid-infrared integrating sphere (A562), which featured a sample port with a diameter of ~2.1 cm and referenced to a diffuse gold standard (Bruker). The infrared beam was used to illuminate on the sample with an incidence angle of 13°. The diffusely reflected part of the infrared radiation is scattered inside the sphere and detected as the reflected luminous flux after integration by a detector located behind the outlet port.

The transmittance and total reflectance spectra of the samples at 300–2500 nm were measured using an ultraviolet–visible–near-infrared (UV–Vis–NIR) spectrophotometer (Agilent Cary 5000) equipped with a diffuse integrating sphere (110 mm sphere, port to sphere area ratio: <3%). $BaSO_4$ was used as standard whiteboard. In all cases, the measurements were performed at ~20 °C and the relative humidity was maintained at ~45%.

The samples were photographed and filmed using a digital camera (Nikon D3100). The samples were placed on a hot plate at 40 °C. An infrared thermal imager (FLIR SC7300M) with a working range of 3.7–4.8 μm was used to capture the MWIR images (the predefined emissivity was set to 1). LWIR images and movies were recorded using an FLIR T1050sc infrared thermal imager with a working range of 7.5–14 μm (the predefined emissivity was set to 0.95 with a focal length

of 36 mm and F number of 1.15). The measurements were performed at ~20 °C, and the relative humidity was maintained at ~45%. FLIR Tools V 5.6 and FLIR ResearchIR Max 4.0 software were used to analyze the LWIR movie data. The apparent temperature of the samples was extracted after averaging using the box measurement tool in the FLIR software packages.

The Al content of the AZO NCs was determined using inductively coupled plasma optical emission spectrometry (ICP-OES, Agilent 720ES). The relative error in the extracted Al content was reduced by averaging six replicate measurements for each batch of AZO NCs.

The surface morphologies of the AZO NCs films were examined using field-emission scanning electron microscopy (FE-SEM; TESCAN MIRA) with a beam energy of 20 keV. Low- and high-resolution transmission electron microscopy (TEM) measurements were carried out using an FEI Talos F200S instrument. To prepare the samples for TEM analysis, a drop of a toluene solution containing the NCs was dried on the surface of an ultrathin carbon film on a copper grid. The cross-section of a $BaF_2$/AZO NC film/ITO film half-device was prepared using a focused ion beam (FIB; FEI STRATA 400 S). The cross-sectional morphologies were then studied using TEM (FEI Talos F200s) and energy-dispersive spectroscopy (EDS; super-X).

Various voltages (−2.5 to 2.5 V) were applied to the AZO nanocrystalline films for 10 min using 1 M LiTFSI as the electrolyte and saturated Ag/AgCl as the counter electrode. After the electrolyte was removed by wiping the sample clean, the samples were sent to the X-ray photoelectron spectroscopy (XPS) sample chamber for quasi-in-situ XPS analysis. XPS was performed using a Thermo ESCALAB 250Xi system with a monochromatic Al $K_\alpha$ source (1486.6 eV). The thickness of the AZO NC film was measured using a Bruker Dektak XT Profiler.

A PARSTAT 4000 Advanced Electrochemical System (Princeton Applied Research, USA) was used to perform cyclic voltammetry (CV) on the AZO NC films and evaluate the cycling performance of the TDIE regulator. In the device performance demonstration, a DC-stabilized power supply (UTP1306-II) was used.

## Data availability

All data generated in this study are presented in the paper and/or the Supplementary information. Source data are provided with this paper.

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

## Acknowledgements

We thank E.D. Gaspera (RMIT University, Australia), M.Y.Li, M.Zu, Z. Wang (National University of Defense Technology) for their valuable advice. We thank H.Z. Yao (Huanghuai University) for assistance with the COMSOL modeling simulation. We also thank Y. Chen (National University of Defense Technology) for assistance with the Profiler analyses. We acknowledge financial support from the National Natural Science Foundation of China (52073303, D.L.), the Natural Science Foundation of Hunan Province (2021JJ10049, D.L.), and Postgraduate Scientific Research Innovation Project of Hunan Province (CX20210054, Y.J.).

## Author contributions

D.L. conceived the idea. Y.J. made further innovations. D.L. and H.C. supervised the work and guided the project. Y.J. fabricated the TDIE regulators, conducted the electrical and spectral characterizations, developed the optical modeling, and analyzed the application of TDIE regulators under D.L.'s supervision. Y.J. and D.C. synthesized the AZO NCs under Y. Jin's supervision. T.L., Z.M., and B.C. fabricated White/Black devices and smart glass. J.T. designed the circuit and wrote the program of infrared displays. S.Z. and X.W. assisted in the analysis of the mechanism of infrared emissivity regulation. C.C. participated in data analysis. Y.J. and D.L. analyzed the data and wrote the manuscript. All the authors discussed the results and commented on the manuscript.

## Competing interests

D.L., Y.J., and H.C. are listed as inventors on five patent applications related to this work filed by the National University of Defense Technology to the National Intellectual Property Administration of P. R. China (application nos. 202310131472X, 2023101314734, 2023101314749, 2023101314787, and 2023101314791; application date: 17 February 2023). The authors declare no competing interests.

## Additional information

**Supplementary information** The online version contains
supplementary material available at

Dongqing Liu or Haifeng Cheng.

**Peer review information** *Nature Communications* thanks Gil Ju Lee and
the other, anonymous, reviewer(s) for their contribution to the peer
review of this work. A peer review file is available.

