## [Peer Review File · Nature Communications]

Transparent dynamic infrared emissivity regulatorsREVIEWER COMMENTS

Reviewer #1 (Remarks to the Author):

This work investigates a transparent dynamic infrared emissivity (TDIE) regulator based on engineered LSPR peaks with AZO NCs. The TDIE is easily controlled by electrical voltage and exhibits a significant emissivity contrast. I believe this work presents a novel contribution to the field of TDIE regulators, but I have some concerns:

1. The authors claim that the proposed TDIE possesses a long cycle life, but it might not be sufficient for smart window applications, which the authors identify as the main application of TDIE. For smart window applications, the authors should demonstrate the long-term operation of TDIE in outdoor environments.
2. The authors mention that solar opacity decreases the efficiency of spectrum utilization when considering smart window applications. However, in my opinion, regulating the solar spectrum is more effective for smart thermal management of buildings than regulating the LWIR region due to a considerable mismatch between radiative and solar absorbed powers. Thus, the authors should clarify the purpose of the TDIE. I believe the TDIE proposed by the authors is useful for smart window applications, not smart thermal management of buildings, as buildings consist of exteriors and windows.
3. In the introduction, the authors discuss the characteristics of electron injection/extraction, but the explanation is somewhat difficult to understand with only text. I suggest adding a brief supplementary figure to clarify the content.
4. Regarding the sentence, "The infrared absorption peaks in Fig. 1c are attributed to the infrared absorption of the LiTFSI/tetraglyme electrolyte in the fingerprint region," why does the absorption peak exist? The electrolyte is beneath the ITO, and ITO is an IR-reflective material. The authors should clarify this part.
5. The emissivity contrast in this paper ranges from ~ 0.4 to 0.51, depending on the wavelength. It appears that a larger emissivity difference would lead to better performance. Could the authors provide options for enhancing the emissivity contrast using simulation or computational studies?
6. In terms of long cycle performance, stable operation over long cycles is crucial, but durability in outdoor environments is even more important. Have you studied outdoor durability? For instance, measuring temperature over several days by changing voltage to match ambient air temperature.
7. Was Figure 2d conducted outdoors? Please clarify the experimental conditions.
8. Figure 5a presents crucial data for smart window applications of TDIE. However, the authors only provide simulation results. I believe an outdoor experiment must be conducted to evaluate the performance of TDIE as a smart window. Specifically, since the authors mention smart window applications, they should prepare an enclosure with the TDIE applied as a window and measure the indoor temperature of the enclosure. Furthermore, the authors should compare the performance of TDIE with other related research (Advanced Optical Materials 9 (13), 2002226/Science Advances 6 (36), eabb1906/ Solar Energy Materials and Solar Cells 230, 111173/ APL Photonics 8 (3), 030801).

Reviewer #2 (Remarks to the Author):

In this work, the authors developed the transparent dynamic infrared emissivity (TDIE) regulators by combining infrared plasmon regulation of aluminum-doped zinc oxide (AZO) nanocrystals with the structural design of infrared reflection indium tin oxide (ITO) electrodes, achieving independent modulation of the infrared emissivity while maintaining a high visible transparency (84.7% at 400-760 nm). Device structure, regulation mechanism, application demonstration, and performance of TDIE regulators were analyzed in detail. Generally speaking, this paper provides significant substantial novelty, and the content is rich. In my opinion, this manuscript can be recommended for minor revision based on following comments:

[1] The author needs to provide infrared transmittance data of TDIE regulators, corresponding to Figure 1c. The authors mentioned in application demonstration that building energy savings were achieved by stacking TDIE regulators on white equipment. In this case, high infrared transmittance

modulation seems to be more advantageous.

[2] The authors claimed that TDIE regulators can achieve an infrared radiated power modulation of 64.9 W/m² in the 7.5-13 μm infrared atmospheric window at 298 K, but there is no experimental data in the manuscript to support this. Therefore, the authors' assessment of average annual HVAC energy savings is not credible. The outdoor test is necessary.

[3] The authors need to evaluate the impact of environmental factors on TDIE regulators performance, including humidity, dust, rainfall, etc.

[4] Some recent reviews (10.1002/smll.202100446; 10.1016/j.nanoen.2022.107435) on dynamic radiation modulation are still missing.

Reviewer #3 (Remarks to the Author):

I have reviewed the manuscript titled "Transparent dynamic infrared emissivity regulators" submitted by Yan Jia et al. and found it to be a well-written and original study that demonstrates the modulation of infrared emissivity based on infrared plasmon regulation of constituent AZO nanocrystals, while maintaining high visible transparency.

I believe this research has the potential to open new opportunities in thermal management schemes, including radiative cooling, which has great potential for reducing energy consumption and mitigating urban heat island effects. Therefore, I recommend acceptance of this study, but would like to suggest some minor revisions.

- While the DIE device shows a high transmittance of nearly 85% in the visible range, it would be necessary to analyze the haze level of the AZO layer, composed of nanoparticles, which may not be entirely free from scattering effects.

- An explanation for the observed difference in the modulation trends of emissivity according to the applied voltage for electron injection and extraction to/from the AZO NP in Figure 2a, b, and Supporting Figure 7 would be helpful.

- The authors have suggested that capacitive charging/discharging of AZO NP affecting its LSPR characteristics is a major cause of the emissivity regulation of the TDIE regulator. It would be useful to provide an explanation for the range of surface potential that was set from 0.8 eV to 2.0 eV.

- It would be beneficial to identify which constituent layer is causing the increase in response time as the number of cycles increases and suggest a way to address it. Additionally, stability over extended operation time is a crucial criterion for thermal management purposes, and I would appreciate information on whether the emissivity characteristics of the TDIE regulator are stable at the same applied voltage over an extended time.

- Lastly, I noticed a few minor errors in several places, such as "Visiable" in Figure 1c. Therefore, I kindly request you to double-check the manuscript thoroughly.

Dear Reviewers:

Re: NCOMMS-23-14432

Thank you for your comments concerning our manuscript entitled “Transparent dynamic infrared emissivity regulators”. Those comments are all of great value and provide significant assistance in revising and enhancing our paper, as well as serving as important guidance for our research endeavors.

We have studied the comments carefully and **revisions in the text are shown using yellow highlight**. The main corrections in the paper and the responds to the reviewer’s comments are listed as followings.

Thank you again your comments.

Responds to the reviewer's comments:

Responses to Reviewer #1:

This work investigates a transparent dynamic infrared emissivity (TDIE) regulator based on engineered LSPR peaks with AZO NCs. The TDIE is easily controlled by electrical voltage and exhibits a significant emissivity contrast. I believe this work presents a novel contribution to the field of TDIE regulators.

Comment 1:

1. The authors claim that the proposed TDIE possesses a long cycle life, but it might not be sufficient for smart window applications, which the authors identify as the main application of TDIE. For smart window applications, the authors should demonstrate the long-term operation of TDIE in outdoor environments.

Revision 1:

Thank you for your suggestions. As you mentioned, the long-term operation of TDIE regulators outdoors is crucial for their application.

We set up an experimental device outdoors, shown in Fig. 1. We simulated the long-term operation of the TDIE regulator outdoors. The infrared thermal imager was used to record the apparent temperature of TDIE regulator under positive/negative operating voltages. TDIE regulator can operate stably under outdoor environment for over 6 days, shown in Fig. 2. During long-term operation of TDIE regulator, its visible light

transmittance decreased to 58.9% (Fig. 3). This is because after applying voltage for a long time, Li^+ in the LiTFSI based electrolyte is reduced to metal Li and reacts with O_2 , N_2 , and water vapor in the air to form Li_2O , Li_3N , and LiOH ^[1]. The contact between air and electrolyte may be due to the transparent silicone used in device encapsulation having a certain gas permeability^[2]. In addition, the AZO NC film also partially corroded and peeled off after long-term operation for more than 7 days (Fig. 4), which was due to the corrosion of the LiTFSI electrolyte^[3, 4]. In future work, we will further improve the long-term operating stability of TDIE regulator by using less corrosive and more stable electrolytes and adopting more advanced packaging processes. We have added this experimental work in the manuscript and supplementary Note 4.

The long-term outdoor operation stability is crucial for the application of TDIE regulators. TDIE regulator can operate stably under outdoor environment for over 6 days, shown in supplementary Note 4. During long-term outdoor operation of the TDIE regulator, the visible light transmittance of the TDIE regulator decreased to 58.9%, and the AZO nanocrystal film was partially corroded and peeled off due to the corrosion and oxidation of the electrolyte (Supplementary Fig. 19, Supplementary Fig. 20). More stable and less corrosive electrolytes will be adopted in future research work.

Fig. 1 The test setup for TDIE regulator in outdoor environments.

Fig. 2 The long-term outdoor operation of TDIE regulator (Date: 2023.5.23 to 2023.5.29; Location: Changsha, China).

Fig. 3 (a) the optical photo of TDIE regulator after long-term outdoor operation; (b) the transmittance of TDIE regulator after long-term outdoor operation.

Fig. 4 Infrared image of AZO NC film peeling after long-term operation of TDIE regulator.

[1] WU X, DU Z J E C. Study of the corrosion behavior of LiFSI based electrolyte for Li-ion cells [J]. *Electrochemistry Communications*, 2021, 129:107088.

[2] ZHANG H, CLOUD A. The permeability characteristics of silicone rubber [J]. *Global Advances in Materials Process Engineering*, 2006, 72-75.

[3] LUO C, LI Y, SUN W, et al. Revisiting the corrosion

mechanism of LiFSI based electrolytes in lithium metal batteries [J].
Electrochimica Acta, 2022, 419:140353.

[4] MATSUMOTO K, INOUE K, NAKAHARA K, et al.
Suppression of aluminum corrosion by using high concentration LiTFSI
electrolyte [J]. Journal of power sources, 2013, 231:234-238.

Comment 2:

2. The authors mention that solar opacity decreases the efficiency of spectrum utilization when considering smart window applications. However, in my opinion, regulating the solar spectrum is more effective for smart thermal management of buildings than regulating the LWIR region due to a considerable mismatch between radiative and solar absorbed powers. Thus, the authors should clarify the purpose of the TDIE. I believe the TDIE proposed by the authors is useful for smart window applications, not smart thermal management of buildings, as buildings consist of exteriors and windows.

Revision 2:

Thank you for your suggestion. As you mentioned, regulating the solar spectrum is a more effective methods than regulating the infrared spectrum, because the energy in the solar spectrum is higher than the energy radiated by objects at ambient temperature. The function of TDIE regulator is mainly that it allows the smart window to achieve independent

regulation of a wide spectrum, from visible light to infrared band. For example, the traditional WO_3 smart window can regulate the visible band of the solar spectrum (transparent to blue color). If the TDIE regulator and the WO_3 smart window were combined used, the spectrum regulation range of the smart window can be expanded, from visible light to infrared band. The independent regulation of a broad spectrum can enhance the ability of smart windows to meet diverse thermal management requirements. We have revised the content in the manuscript to clarify the purpose of the TDIE.

For smart thermal management of smart windows, it is necessary to simultaneously achieve independent regulation of solar spectral and infrared emissivity to better meet diverse thermal control requirements.

Highly transparent DIE regulators are anticipated as they can be placed on top of a solar-spectrum dynamic regulation device or visible color-changing device to achieve independent dynamic regulation of the infrared emissivity and solar spectrum.

Comment 3:

3. In the introduction, the authors discuss the characteristics of electron injection/extraction, but the explanation is somewhat difficult to understand with only text. I suggest adding a brief supplementary figure to clarify the content.

Revision 3:

Thank you for your suggestion. We have added a supplementary diagram of electron injection/extraction in the manuscript and supplementary materials to facilitate understanding of the content. Thank you again for your suggestion.

When the oxidation potential ($E_{\text{surf}} = 0.8 \text{ eV}$) is applied, electrons were extracted from AZO NC (Supplementary Fig. 27). The width of the surface depletion layer (W_{sd}) of an AZO NC is $\sim 1.27 \text{ nm}$, occupying 54.3% of the NC volume (Fig. 3d). Electrons were injected into the AZO NC as the surface potential increased, resulting in a decrease in W_{sd} (Supplementary Fig. 27).

Supplementary Fig. 27 Schematic diagram of electron injection/extraction in the surface depletion layer of AZO nanocrystals.

Comment 4:

4. Regarding the sentence, "The infrared absorption peaks in Fig. 1c are attributed to the infrared absorption of the LiTFSI/tetraglyme

electrolyte in the fingerprint region," why does the absorption peak exist? The electrolyte is beneath the ITO, and ITO is an IR-reflective material. The authors should clarify this part.

Revision 4:

Thank you for your suggestion. The absorption peak position of the TDIE regulator after applying a voltage of 2.5 V is the same as that of the LiTFSI/tetraglyme electrolyte absorption peak (Fig. 5). Therefore, the infrared absorption peaks are attributed to the infrared absorption of the LiTFSI/tetraglyme electrolyte in the fingerprint region.

The existence of the absorption peak is due to the fact that ITO is not a dense film, and there are some cracks in the ITO film (Fig. 6). These cracks allow the electrolyte to partially penetrate, thus exhibiting absorption peaks of the electrolyte in the infrared spectrum. Since the area occupied by the cracks is very small, it will not affect the infrared reflection of the ITO film. We have clarified this part in the manuscript.

The infrared absorption peaks in Fig. 1c are attributed to the infrared absorption of the LiTFSI/tetraglyme electrolyte in the fingerprint region.

The existence of the absorption peaks is due to the presence of cracks in the ITO film, allowing the electrolyte to partially penetrate.

Fig. 5 The infrared absorption spectrum of the LiTFSI/tetraglyme electrolyte and the infrared reflection spectrum of the TDIE regulator after applying a voltage of 2.5 V.

Fig. 6 (a) TEM morphology of AZO NC film and ITO film. (b, c) Cracks in the ITO film.

Comment 5:

5. The emissivity contrast in this paper ranges from ~0.4 to 0.51, depending on the wavelength. It appears that a larger emissivity difference

would lead to better performance. Could the authors provide options for enhancing the emissivity contrast using simulation or computational studies?

Revision 5:

Thank you for your suggestion. As you mentioned, a large emissivity contrast leads to better performance. In our research work, we optimized the emissivity contrast through experiments. The functional layers of variable emissivity in the TDIE regulator are AZO nanocrystal layer and ITO infrared reflective layer. The AZO nanocrystal layer can change its infrared LSPR absorption by applying different voltages. The ITO film serves as an infrared reflective background. The size of the AZO nanocrystals, the AZO NC film thickness, and the infrared reflectivity of the ITO thin film have an influence on the emissivity contrast of the TDIE modulator.

The higher the infrared reflectivity of the ITO film, the better the emissivity contrast performance of the TDIE regulator. The infrared reflectivity of the ITO film is determined by the evaporation process, and the optimized infrared reflectivity spectrum is shown in the Fig. 7.

The size of the AZO nanocrystals and the AZO NC film thickness has an important influence on the emissivity contrast because they can affect the LSPR absorption of AZO nanocrystals before and after electron injection/extraction, thus affecting the emissivity contrast of the TDIE

regulator. We synthesized AZO nanocrystals with different sizes, as shown in Fig. 8 (a-d). We investigated the effect of nanocrystal size on emissivity contrast, as shown in Fig. 8 (d, e). There is an optimal AZO NC size (diameter: ~ 11 nm) around which the emissivity contrast is greatest. We also studied the emissivity contrast of TDIE regulators prepared by AZO nanocrystals with different film thicknesses. The result, as shown in the Fig. 9. There is an optimal film thickness (~ 1.2 μm) around which the emissivity contrast is greatest.

Through experiments, we optimized the preparation parameters of the TDIE regulator to maximize the emissivity contrast. We have tried simulation or calculation methods to optimize the emissivity contrast. However, accurately obtaining the dielectric constant of individual nanocrystals is challenging. When the nanocrystals are in contact, they will undergo coupling which has an impact on the dielectric constant. Additionally, modeling films with randomly distributed nanocrystals remains limited in accuracy. Optimizing emissivity contrast by experiment has better reliability and is easier to implement. We thus conducted experimental optimization of the emissivity contrast. Thank you again for your suggestion.

Fig. 7 The infrared reflectivity of optimized ITO film.

Fig. 8 a-d, Morphology of AZO nanocrystals with different sizes. e and f, The effect of AZO nanocrystal size on the emissivity contrast of 3-5 μm and 7.5-13 μm.

Fig. 9 The effect of AZO NC film thickness on the emissivity contrast of 3-5 μm and 7.5-13 μm .

Comment 6:

6. In terms of long cycle performance, stable operation over long cycles is crucial, but durability in outdoor environments is even more important. Have you studied outdoor durability? For instance, measuring temperature over several days by changing voltage to match ambient air temperature.

Revision 6:

Thank you for your suggestion. The durability of outdoor environments is crucial for the application of TDIE regulator. We conducted outdoor durability experiments in Revision 1. TDIE regulator can operate stably under outdoor environment for over 6 days, shown in Fig. 2. However, the corrosiveness and stability of the electrolyte and the packaging process of the device limit the further use of TDIE regulator. In future work, we will further improve the long-term operating stability of

TDIE regulator by using less corrosive and more stable electrolytes and adopting more advanced packaging processes. We have added this experimental work in the manuscript and supplementary Note 4. Thank you again for your suggestion.

The long-term outdoor operation stability is crucial for the application of TDIE regulators. TDIE regulator can operate stably under outdoor environment for over 6 days, shown in supplementary Note 4. During long-term outdoor operation of the TDIE regulator, the visible light transmittance of the TDIE regulator decreased to 58.9%, and the AZO nanocrystal film was partially corroded and peeled off due to the corrosion and oxidation of the electrolyte (Supplementary Fig. 19, Supplementary Fig. 20). More stable and less corrosive electrolytes will be adopted in future research work.

Comment 7:

7. Was Figure 2d conducted outdoors? Please clarify the experimental conditions.

Revision 7:

The cycle test of Figure 2d was conducted in the indoor laboratory environment. We have added its experimental conditions in the main text. Thank you for your suggestion.

Fig. 2 d, High stability of the maximum and minimum apparent

temperatures of LWIR during cycling, indicating the good endurance of the TDIE regulators. The test was conducted in the indoor laboratory environment.

Comment 8:

8. Figure 5a presents crucial data for smart window applications of TDIE. However, the authors only provide simulation results. I believe an outdoor experiment must be conducted to evaluate the performance of TDIE as a smart window. Specifically, since the authors mention smart window applications, they should prepare an enclosure with the TDIE applied as a window and measure the indoor temperature of the enclosure. Furthermore, the authors should compare the performance of TDIE with other related research (Advanced Optical Materials 9 (13), 2002226/Science Advances 6 (36), eabb1906/ Solar Energy Materials and Solar Cells 230, 111173/ APL Photonics 8 (3), 030801).

Revision 8:

Thank you for your suggestion. We conducted an outdoor experiment to evaluate the performance of TDIE regulator as a smart window. As you suggested, we prepared an enclosure chamber and used TDIE regulator applied as a window. We measured the indoor temperature change of the chamber depend on the emissivity of TDIE regulator. The schematic diagram of the experimental setup is shown in Fig. 10. A chamber is dug

out of the polystyrene foam box, and then the foam box is wrapped with Al foil. The TDIE regulator is covered above the chamber as a window. The chamber was sealed using a low-density polyethylene (PE) film as a convection shield to reduce the effect of thermal convection on the sample. The optical photo of experimental setup is shown in Fig. 11. Since solar radiation has a significant effect on chamber temperature, we chose to conduct experiments at nighttime for investigating the impact of infrared radiation regulation of TDIE devices on chamber temperature. A control sample with a fixed emissivity (0.4 in 8-14 μm) is covered above the same chamber for comparison. In order to eliminate the influence of outdoor environmental temperature fluctuations over time, we calculated the temperature difference (ΔT) between the chamber temperature covered with TDIE regulator and the environment temperature. The change in ΔT is attributed to the regulation of infrared emissivity by the TDIE regulator. The average temperature change of the chamber is 0.61 $^{\circ}\text{C}$, achieved by regulating infrared emissivity of TDIE regulator (Fig. 12). The ΔT between the chamber temperature covered with the fixed emissivity (0.4 in 8-14 μm) sample and the environment temperature is almost constant (Fig. 13), which proves that the change of ΔT is due to the infrared emissivity regulation. In conclusion, we have proved that the temperature of the enclosure chamber can be regulated by regulating the emissivity of the TDIE regulator, and the temperature regulation range is ~ 0.61 $^{\circ}\text{C}$. We have

added this experimental work in the manuscript and Supplementary Note 6.

We have also cited the related research as you mentioned in the manuscript and compared the performance of the TDIE regulator with theirs. Thank you again for your suggestion.

The chamber using the TDIE regulator can achieve a temperature regulation of 0.61 °C by regulating the emissivity of the TDIE regulator (Supplementary Note 6). Compared with the radiative coolers³²⁻³⁵, TDIE regulators can achieve an infrared radiated power modulation of 23.1 W/m².

32. Kim, M. et al. Visibly Transparent Radiative Cooler under Direct Sunlight. *Advanced Optical Materials* **9**, 2002226 (2021).

33. Heo, S. Y. et al. A Janus emitter for passive heat release from enclosures. *Science advances* **6**, eabb1906 (2020).

34. Lee, G. J., Heo, S. Y., Kang, I. S. & Song, Y. M. Thermostat property of janus emitter in enclosures. *Solar Energy Materials Solar Cells* **230**, 111173 (2021).

35. Kim, D. H. et al. Polarization-mediated multi-state infrared system for fine temperature regulation. *APL Photonics* **8**, 030801 (2023).

Fig. 10 Schematic of the experimental setup. TDIE regulator was applied as a window of an enclosure chamber and the indoor temperature of the chamber was measured.

Fig. 11 The optical photos of the experimental setup.

Fig. 12 The temperature difference (ΔT) between the chamber temperature covered with TDIE regulator and the environment temperature. The voltage (± 2.5 V) of the TDIE regulator is switched every half hour.

Fig. 13 The temperature difference (ΔT) between the chamber temperature covered with control sample (fixed emissivity: 0.4) and the environment temperature.

Responses to Reviewer #2:

In this work, the authors developed the transparent dynamic infrared emissivity (TDIE) regulators by combining infrared plasmon regulation of aluminum-doped zinc oxide (AZO) nanocrystals with the structural design of infrared reflection indium tin oxide (ITO) electrodes, achieving independent modulation of the infrared emissivity while maintaining a high visible transparency (84.7% at 400-760 nm). Device structure, regulation mechanism, application demonstration, and performance of TDIE regulators were analyzed in detail. Generally speaking, this paper provides significant substantial novelty, and the content is rich. In my opinion, this manuscript can be recommended for minor revision based on following comments:

Comment 1:

[1] The author needs to provide infrared transmittance data of TDIE regulators, corresponding to Figure 1c. The authors mentioned in application demonstration that building energy savings were achieved by stacking TDIE regulators on white equipment. In this case, high infrared transmittance modulation seems to be more advantageous.

Revision 1:

Thank you for your suggestion. The TDIE regulator is infrared opaque due to the high infrared reflection characteristics of ITO films. The infrared transmittance curve of the TDIE regulator is shown in the Fig. 14. When

negative voltage is applied to TDIE regulator, electrons are injected into the AZO NCs, the infrared LSPR absorption of AZO nanocrystals is increased. The TDIE regulator show high emissivity state. When a positive voltage is applied to the TDIE regulator, electrons are extracted from AZO nanocrystals. The infrared LSPR absorption of AZO nanocrystals weakens, resulting in an increase in infrared transmittance. TDIE regulator exhibits the high infrared reflectivity characteristics of ITO films, that is, the low emissivity state. Therefore, TDIE regulator is infrared opaque.

We stack the TDIE regulator on top of the black/white devices so that the TDIE regulator can regulate the infrared spectrum and the black/white device can regulate the visible light spectrum. Therefore, the composite device consisting of the TDIE regulator and black/white device can achieve independent regulation over a wide spectral range from visible to infrared.

We have added the infrared transmittance curve of TDIE regulator in Supplementary Fig. 6. Thank you again for your suggestion.

Fig. 14 The infrared transmittance curve (2.5-15 μm) of the TDIE regulator.

Comment 2:

[2] The authors claimed that TDIE regulators can achieve an infrared radiated power modulation of 64.9 W/m^2 in the 7.5-13 μm infrared atmospheric window at 298 K, but there is no experimental data in the manuscript to support this. Therefore, the authors' assessment of average annual HVAC energy savings is not credible. The outdoor test is necessary.

Revision 2:

Thank you for your suggestion. We conducted an outdoor experiment to evaluate the infrared radiated power modulation performance of TDIE regulator. We measure the infrared radiated power modulation amount of TDIE regulator by power compensation method, refer to the previous

literature (RAMAN et al. Passive radiative cooling below ambient air temperature under direct sunlight [J]. *Nature*, 2014, 515(7528): 540-544. doi:10.1038/nature13883).

The schematic diagram of the experimental setup is shown in Fig. 15. The heating sheet is adhered to the back of the TDIE regulator. The thermocouple is sandwiched between the TDIE regulator and the heating sheet for temperature measurement. The chamber was sealed using a low-density polyethylene (PE) film as a convection shield to reduce the effect of thermal convection on the sample. The optical photo of experimental setup is shown in Fig. 16. The experimental setup was built on the roof. Since solar radiation has a significant effect on chamber temperature, we chose to conduct experiments at nighttime for investigating the infrared radiated power modulation of TDIE regulator. Two identical TDIE regulators, combined with heating sheet and thermocouples, were placed in the foam chamber. Before applying voltage to the TDIE regulator (21:15 to 21:25), the temperatures of the two TDIE regulators are basically equal (Fig. 17). Two TDIE regulators are applied with voltages of +2.5V and -2.5V respectively (after 21:25), hereinafter named as TDIE(+) and TDIE(-), respectively. TDIE(+) and TDIE(-) showed a temperature difference of 0.62 °C caused by different emissivity of TDIE(+) and TDIE(-) (Fig. 17). We applied a stepwise increase in heat input to the TDIE(-) using a heating sheet. The temperature of TDIE(-) increases gradually with the increase of

heat input. When the heat input increased to 23.1 W/m^2 , the temperature of TDIE(-) was equal to that of TDIE(+), indicating that the experimental radiated power modulation amount of the TDIE regulator is $\sim 23.1 \text{ W/m}^2$ (Fig. 17).

The experimental radiated power modulation amount of the TDIE regulator is lower than the calculated radiated power modulation amount. This is because the calculated radiated power modulation amount is obtained by the equation S5 and equation S6 (shown below) in the supplementary material. The $\varepsilon(\lambda_1, \lambda_2)$ of TDIE regulator is calculated by the emissivity curve of TDIE regulator. The calculated result is the theoretical maximum radiated power modulation of the TDIE regulator before and after the change of emissivity, without considering the effects of convection and conduction. During the experimental test, the temperature difference caused by the different emissivity of TDIE(+) and TDIE(-) will be influenced by air convection and heat conduction between the TDIE regulator and heating sheet as well as the environment, resulting in the reduction of the experimental radiated power modulation amount of the TDIE regulator.

We have added this outdoor radiated power modulation test in the supplementary Note 6. We supplemented the experimental radiation power modulation and analyzed the reasons for the difference between the experimental and calculated radiated power modulation amount in the

manuscript.

We tested the radiated power regulation of the TDIE regulator outdoors (Supplementary Note 6). Compared with the radiative coolers³²⁻³⁵, the TDIE regulator can achieve a 23.1 W/m² infrared radiated power regulation. The experimental infrared radiated power regulation amount is lower than the calculated value (64.9 W/m² calculated by equation S5 and S6). This is because the calculated radiated power solely relies on the emissivity spectrum of TDIE regulator, whereas experimental radiated power regulation is affected by heat conduction and convection.

The infrared radiation power of the TDIE regulator in an infrared atmospheric window at T can be calculated as

$$P(\lambda_1, \lambda_2) = \varepsilon(\lambda_1, \lambda_2) \times \int_{\lambda_1}^{\lambda_2} I_{BB}(T, \lambda) d\lambda \quad (\text{S5})$$

The change in the infrared radiation power (ΔP) of the TDIE regulator in different emissivity states was computed as

$$\Delta P(\lambda_1, \lambda_2) = P_h(\lambda_1, \lambda_2) - P_l(\lambda_1, \lambda_2) \quad (\text{S6})$$

where $P_h(\lambda_1, \lambda_2)$ and $P_l(\lambda_1, \lambda_2)$ represent the infrared radiation power of the TDIE regulator in the high-and low-emissivity states, respectively.

Fig. 15 Schematic of the experimental setup. The heating plate is adhered to the back of the TDIE regulator. The thermocouple is sandwiched between the TDIE regulator and the heating plate for temperature measurement. PE film serves as a convection shield to reduce the effect of thermal convection on the sample.

Fig. 16 The optical photos of the experimental setup in roof. The power supply is used to provide the voltage for the TDIE regulator and provide energy for the heating sheet for heat input.

Fig. 17 Temperature measurement of two TDIE regulators under applied voltage of 2.5 V and -2.5 V, respectively. The TDIE(-) input a stepwise increasing of heat from 22:00 (as indicated by the quantity displayed on the top of the graph).

Comment 3:

[3] The authors need to evaluate the impact of environmental factors on TDIE regulators performance, including humidity, dust, rainfall, etc.

Revision 3:

Thank you for your suggestion. We simulated the effect of dust covering on the TDIE regulator's infrared emissivity regulation performance (Fig. 18). Dust is a high emissivity material. When a thin layer of dust covers the surface of the TDIE regulator and a voltage of +2.5 V is applied, the dust covered area exhibits high emissivity characteristic due to the presence of dust while the dust uncovered area shows low emissivity characteristic of the TDIE regulator (Fig. 18 (b)). When a thick layer of

dust accumulated on the surface of the TDIE regulator, regardless of whether a positive or negative voltage is applied, TDIE regulator exhibited high emissivity characteristic of dust (Fig. 18 (c)).

We operated the TDIE regulator in an outdoor environment for a week to investigate the influence of environmental factors (humidity, rainfall, wind, etc.) on TDIE performance. We set up an experimental device outdoors, shown in Fig. 19. The outdoor temperature and humidity were recorded by hygromograph. Wind speed was recorded by anemograph. The weather conditions for these 7 days are shown in Table 1.

On rainy days, raindrops will fall on the TDIE regulator (Fig. 20). Since water is an infrared-opaque high-emissivity state, when the water film is covered on the TDIE regulator, the TDIE regulator will exhibit the high emissivity of the water film (Fig. 20). When the water film evaporates, the TDIE regulator can resume emissivity regulation (Fig. 21). The temperature, humidity, and wind speed are showed in Fig. 22, Fig. 23, and Fig. 24, respectively. Humidity and wind speed have no significant impact on the TDIE regulator. During the 7 days of testing, the TDIE regulator can operate stably in different environmental factors (Fig. 25).

We have added the impact of environmental factors on the TDIE regulator in Supplementary Note 4. We analyzed the effects of humidity, dust, and rainfall on the TDIE regulator in the manuscript. Thank you for your suggestion again.

We investigated the impact of environmental factors on the performance of TDIE regulators. The air humidity has no significant impact on the TDIE regulator. If dust and rain fall on the TDIE regulator, the TDIE regulator will show high emissivity characteristics of dust and water (Supplementary Fig. 21, Supplementary Fig. 22).

Fig. 18 **a**, optical and infrared photos of the TDIE regulator without dust; **b**, optical and infrared photos of the TDIE regulator with a thin layer of dust on its surface; **c**, optical and infrared photos of the TDIE regulator with a thick layer of dust on its surface.

Fig. 19 The test setup for TDIE regulator in outdoor environments.

Table 1 The humidity, temperature, weather, and PM2.5 of Changsha (Date: 2023.5.23 to 2023.5.29).

Date	Humidity (%)	Temperature (°C)	Weather	PM2.5 ($\mu\text{g}/\text{m}^3$)
2023.5.23	57-92	15-22	Light rain	53
2023.5.24	37-92	17-26	Cloudy	39
2023.5.25	52-77	19-30	Cloudy	36
2023.5.26	44-88	24-31	Moderate rain	37
2023.5.27	43-88	25-35	Sunny	48
2023.5.28	39-68	25-35	Sunny	28
2023.5.29	47-86	25-36	Cloudy	58

Fig. 20 Optical and infrared photos of rainwater falling on the TDIE regulator.

Fig. 21 Infrared photos of TDIE regulator applying -2.5 V and 2.5 V voltage after water film evaporation.

Fig. 22 Environment temperature measured by hygrothermograph (Date: 2023.5.23 to 2023.5.29; Location: Changsha, China).

Fig. 23 Humidity measured by hygrothermograph (Date: 2023.5.23 to 2023.5.29; Location: Changsha, China).

Fig. 24 Wind speed measured by anemograph (Date: 2023.5.23 to 2023.5.29; Location: Changsha, China).

Fig. 25 The long-term outdoor operation of TDIE regulator (Date: 2023.5.23 to 2023.5.29; Location: Changsha, China).

Comment 4:

[4] Some recent reviews (10.1002/sml.202100446;

10.1016/j.nanoen.2022.107435) on dynamic radiation modulation are still missing.

Revision 4:

Thank you for your suggestion. We have cited relevant literature in the manuscript.

8. Wei, H. et al. Smart Materials for Dynamic Thermal Radiation Regulation. *Small* **17**, 2100446, (2021).

9. Zhai, H., Fan, D. & Li, Q. Dynamic radiation regulations for thermal comfort. *Nano Energy*, 107435 (2022).

Responses to Reviewer #3:

I have reviewed the manuscript titled "Transparent dynamic infrared emissivity regulators" submitted by Yan Jia et al. and found it to be a well-written and original study that demonstrates the modulation of infrared emissivity based on infrared plasmon regulation of constituent AZO nanocrystals, while maintaining high visible transparency.

I believe this research has the potential to open new opportunities in thermal management schemes, including radiative cooling, which has great potential for reducing energy consumption and mitigating urban heat island effects. Therefore, I recommend acceptance of this study, but would like to suggest some minor revisions.

Comment 1:

1. While the DIE device shows a high transmittance of nearly 85% in the visible range, it would be necessary to analyze the haze level of the AZO layer, composed of nanoparticles, which may not be entirely free from scattering effects.

Revision 1:

Thank you for your suggestion. We tested the haze of AZO layer and TDIE regulator, as shown in Fig. 26. The haze of AZO layer is 2.22% (Fig. 26). The scattering effect generated by AZO nanoparticles is relatively small, because the particle size of AZO nanoparticles (< 20 nm) is much smaller than the wavelength of visible light (400-760 nm). We also tested

the haze of the TDIE regulator, which was 2.11%. The haze of the TDIE regulator is slightly smaller than that of the AZO layer. This is due to the wetting effect of the liquid electrolyte, filling some of the pores, which has also been reported in other literatures^[5].

We have added the haze curve in the Supplementary Fig. 5 and we have analyzed the haze of TDIE regulator in the manuscript. Thank you again for your suggestions.

The haze of the TDIE regulator is small, only 2.11%, due to the low scattering of the AZO NC film and the wetting effect of the liquid electrolyte (Supplementary Fig. 5).

Fig. 26 The haze of AZO layer and TDIE regulator.

[5] MANDAL J, JIA M, OVERVIG A, et al. Porous Polymers with Switchable Optical Transmittance for Optical and Thermal Regulation [J].

Joule, 2019, 3(12): 3088-3099.

Comment 2:

2. An explanation for the observed difference in the modulation trends of emissivity according to the applied voltage for electron injection and extraction to/from the AZO NP in Figure 2a, b, and Supporting Figure 7 would be helpful.

Revision 2:

Thank you for your suggestion. After applying different voltages, the state of charge (SoC) of AZO nanocrystals varies. With the change of the applied voltage, the number of electrons injected into the AZO nanocrystals varies, resulting in the change of the LSPR absorption of the AZO nanocrystals. For example, when a more negative voltage is applied, more electrons are injected into the AZO nanocrystal, and its LSPR absorption is stronger, therefore the AZO nanocrystal applied a more negative voltage shows a higher emissivity (Figure 2a, b, and Supporting Figure 7).

We have added relevant explanations in the manuscript to facilitate understanding.

With the change of the applied voltage, the number of electrons injected into the AZO NCs varies, resulting in the change of the LSPR absorption of the AZO NCs. For example, when a more negative voltage is applied, more electrons are injected into the AZO NC, and its LSPR absorption is

stronger. Therefore, TDIE regulator exhibited different infrared emissivity after applying different voltage (Fig. 2a, Fig. 2b, Supplementary Fig. 9).

Comment 3:

3. The authors have suggested that capacitive charging/discharging of AZO NP affecting its LSPR characteristics is a major cause of the emissivity regulation of the TDIE regulator. It would be useful to provide an explanation for the range of surface potential that was set from 0.8 eV to 2.0 eV.

Revision 3:

Thank you for your suggestion. We provided an explanation for the range of surface potential that was set from 0.8 eV to 2.0 eV in the Supplementary Note 5. Supplementary Fig. 28 shows the band energetics of the AZO NCs used to solve the Poisson's equation. E_{ref} is the reference potential and the center of the band gap; E_{surf} is the applied surface potential relative to E_{ref} ; E_{F} (1.85 eV) is the Fermi level. When the surface potential is equal to the Fermi level (1.85 eV), the energy band does not bend. The potential below the Fermi level (1.85 eV) is the oxidation potential. The minimum oxidation potential was set as 0.8 eV, referring to previous literature^[5]. The potential higher than the Fermi level is the reduction potential, and we set the surface potential of 2 eV as the maximum reduction potential. Higher reduction potentials were not set, since the

results did not converge when solving Poisson's equation for higher reduction potentials. Similar surface potential settings have also been reported in ITO nanoparticles^[6, 7].

We have also provided an explanation in the manuscript for the range of surface potential that was set from 0.8 eV to 2.0 eV.

The surface potential was set from 0.8 eV to 2.0 eV, referring to previous literature¹⁹. Higher potentials above 2.0 eV were not set, since the result of solving Poisson's equation did not converge.

[6] ZANDI O, AGRAWAL A, SHEARER A B, et al. Impacts of surface depletion on the plasmonic properties of doped semiconductor nanocrystals [J]. Nature Materials, 2018, 17(8): 710-717.

[7] AGRAWAL A, KRIEGEL I, RUNNERSTROM E L, et al. Rationalizing the Impact of Surface Depletion on Electrochemical Modulation of Plasmon Resonance Absorption in Metal Oxide Nanocrystals [J]. ACS Photonics, 2018, 5(5): 2044-2050.

Supplementary Fig. 28 a, Band structure of AZO NCs. **b**, Band bending profiles due to the presence of surface depletion layers. The band bending

is defined as $E_F - E_{\text{surf}}$.

Comment 4:

4. It would be beneficial to identify which constituent layer is causing the increase in response time as the number of cycles increases and suggest a way to address it. Additionally, stability over extended operation time is a crucial criterion for thermal management purposes, and I would appreciate information on whether the emissivity characteristics of the TDIE regulator are stable at the same applied voltage over an extended time.

Revision 4:

Thank you for your suggestion. We analyzed the reasons for the increase in response time in the Supplementary Note 3, and proposed measures to improve response time.

As the number of cycles increases, the response time will slightly increase. This is mainly due to the corrosion of the working ITO electrode after 4000 cycles. In addition, after multiple cycles, the adsorption/desorption rates of Li^+ in the electrolyte decrease on the working electrode and counter electrode. To reduce response time after multiple cycles, a promising avenue is the selection of an electrolyte with greater stability, reduced corrosiveness, and faster ion adsorption/desorption rates.

We analyzed the stability of emissivity of TDIE regulator over an extended time at different voltages. We assume that the TDIE regulator undergoes state transitions four times per day, that is, once every 6 hours. The emissivity of the TDIE regulator can remain stable for more than 6 hours under different voltages (Fig. 27). The emissivity of TDIE regulator can remain stable even when the voltage application time is extended to 12 hours (Fig. 28). This indicates that the emissivity of the TDIE regulator is stable at the same applied voltage over an extended time. The reason is as stated in Revision 2, the number of electrons injected into AZO nanocrystals is determined by applied voltages. A constant voltage will cause the LSPR absorption of AZO nanocrystals to remain constant for an extended time. Therefore, the infrared emissivity of the TDIE regulator is stable at the same applied voltage over an extended time.

We have added the emissivity characteristics of the TDIE regulator over an extended time in the Supplementary Fig. 10 and we have analyzed it in the manuscript. Thank you again for your suggestions.

The emissivity of the TDIE regulator can remain stable for more than 6 hours under different voltages (Supplementary Fig. 10).

Fig. 27 The emissivity of the TDIE regulator applied voltage of (a) - 2.5 V, (b) -2 V, (c) -1.5 V, (d) -1 V, (e) +1 V, (f) +2.5 V over an extended time.

Fig. 28 The emissivity of the TDIE regulator applied voltage of (a) -2.5 V, (b) +2.5 V for 1h, 6h, 12h.

Comment 5:

5. Lastly, I noticed a few minor errors in several places, such as "Visiable" in Figure 1c. Therefore, I kindly request you to double-check

the manuscript thoroughly.

Revision 5:

Thank you for your suggestion. We have modified the “Visible” in Figure 1c. We have reviewed the content of the manuscript again. Thank you again for your suggestion.

REVIEWERS' COMMENTS

Reviewer #1 (Remarks to the Author):

All concerns that I raised were addressed by the authors. I would like to recommend the publication in Nature Communication as it is.

Reviewer #2 (Remarks to the Author):

The authors have addressed all my concerns and I now recommend the acceptance of the manuscript for the publication in Nature Communications.

Reviewer #3 (Remarks to the Author):

I have reviewed the revised version of the manuscript, and I believe that the issues raised from the previous round has been resolved. Therefore, I would like to recommend publication of the current version.

Dear Reviewers:

Re: NCOMMS-23-14432A

Thank you for your comments concerning our manuscript entitled “Transparent dynamic infrared emissivity regulators”.

Detailed below are all the specific comments addressed along with our responses.

Responses to Reviewer #1:**Comment:**

All concerns that I raised were addressed by the authors. I would like to recommend the publication in Nature Communication as it is.

Response:

Thank you for your feedback; we are happy to have satisfactorily addressed your comments.

Responses to Reviewer #2:**Comment:**

The authors have addressed all my concerns and I now recommend the acceptance of the manuscript for the publication in Nature Communications.

Response:

We thank you for your feedback and thank you for the positive comments.

Responses to Reviewer #3:**Comment:**

I have reviewed the revised version of the manuscript, and I believe that the issues raised from the previous round has been resolved. Therefore, I would like to recommend publication of the current version.

Response:

We would like to express our gratitude to your help for improving the manuscript.